# Protocol for developing quality assurance measures to use in surgical trials: an example from the ROMIO study

Natalie S Blencowe,[1,2] Anni Skilton,[3] Daisy Gaunt,[4] Rachel Brierley,[5] Andrew Hollowood,[2] Simon Dwerryhouse,[6] Simon Higgs,[6] William Robb,[7] Alex Boddy,[8] George Hanna,[9] C Paul Barham,[10] Jane Blazeby,[1,2] on behalf of the ROMIO Study team

For numbered affiliations see end of article.

**Correspondence to**
Natalie S Blencowe;
natalie.blencowe@bristol.ac.uk

## ABSTRACT

**Introduction** Randomised controlled trials (RCTs) in surgery are frequently criticised because surgeon expertise and standards of surgery are not considered or accounted for during study design. This is particularly true in pragmatic trials (which typically involve multiple centres and surgeons and are based in 'real world' settings), compared with explanatory trials (which are smaller and more tightly controlled).

**Objective** This protocol describes a process to develop and test quality assurance (QA) measures for use within a predominantly pragmatic surgical RCT comparing minimally invasive and open techniques for oesophageal cancer (the NIHR ROMIO study). It builds on methods initiated in the ROMIO pilot RCT.

**Methods and analysis** We have identified three distinct types of QA measure: (i) entry criteria for surgeons, through assessment of operative videos, (ii) standardisation of operative techniques (by establishing minimum key procedural phases) and (iii) monitoring of surgeons during the trial, using intraoperative photography to document key procedural phases and standardising the pathological assessment of specimens. The QA measures will be adapted from the pilot study and tested iteratively, and the video and photo assessment tools will be tested for reliability and validity.

**Ethics and dissemination** Ethics approval was obtained (NRES Committee South West—Frenchay, 25 April 2016, ref: 16/SW/0098). Results of the QA development study will be submitted for publication in a peer-reviewed journal.

**Trial registration number** ISRCTN59036820, ISRCTN10386621.

## Strengths and limitations of this study

► This protocol describes the process of developing quality assurance (QA) measures to use within pragmatic surgical randomised controlled trials, which is an area lacking in methodological guidance.

► Further work will establish if these QA approaches can be used more widely in different contexts.

► This study will explore the feasibility of obtaining digital videos of open surgery during this study; however, it may not be possible to achieve this with good enough quality or without interfering with the process of surgery. The increasing availability and use of digital imaging and information technologies should facilitate the use of these methods.

► QA processes require investment of resource and expertise: future work to streamline them is therefore needed.

► While the addition of QA processes can reduce bias and improve internal validity, they may initially appear to compromise a trial's generalisability. Provided pragmatic standards are set and monitored, this should be avoided.

## INTRODUCTION AND RATIONALE

Randomised controlled trials (RCTs) in surgery are notoriously difficult to design and conduct, due to numerous methodological and cultural challenges. Many of these challenges relate to the fact that surgical procedures are complex healthcare interventions, meaning that 'unlike 20 milligram tablets, no two procedures are the same' and achieving standardisation of surgical techniques and processes is difficult.[1] This is partly because surgeons naturally undertake procedures in (slightly) different ways and have differing skill levels, which may influence rates of postoperative complications and reoperation.[2] A lack of consideration for intervention standardisation and surgeon expertise in the context of RCTs may introduce bias, compromising internal validity. This is acknowledged in guidance such as Consolidated Standards of Reporting Trials of Non-Pharmacologic Treatment (CONSORT-NPT) and Standard Protocol Items: Recommendations for Interventional Trials (SPIRIT), which provides a checklist of 33 items to be reported in trial

protocols. They recommend reporting precise details of the intervention and its components to enable replication in routine practice, information about standardisation, procedures for monitoring adherence to intervention protocols and consideration for the expertise of care providers.

Neither CONSORT-NPT nor SPIRIT differentiate between the information required in pragmatic or explanatory settings. This may be because it is rare for trials to be purely pragmatic or explanatory: trial design is not dichotomous and there is a continuum between the two extremes. In explanatory trials, which determine the efficacy of interventions, great detail may be necessary because the interventions are often novel and their safety needs to be assessed within carefully controlled settings. Pragmatic trials, which determine whether interventions are effective in the real world, are often multicentre studies with large numbers of surgeons.[3] Under such circumstances, specifying each operative step is likely to create difficulties, and ensuring that each step was delivered as planned may be unrealistic. A balance between adequate standardisation and practicality is therefore necessary and appropriate. One way of achieving this is to determine the minimum active ingredients of the intervention—those that are thought to optimise outcomes or those that are different between the interventions in each trial group—and the degree to which they need to be standardised. In this way, monitoring only the key components may be sufficient, rather than monitoring all components and steps, in order to ensure the intervention is actually delivered as planned.[4] This approach would also account for the fact that most trials sit within the pragmatic-explanatory continuum rather than being one or the other.

It is, therefore, important to provide reassurance about the standards of surgery in all RCTs, while recognising that this may vary according to trial design. One way of achieving this is to undertake quality assurance (QA), defined as the process(es) of 'directing the performance and behaviours of practitioners and institutions toward more appropriate and acceptable health outcomes'.[5] Undertaking QA in surgery has been summarised in a systematic review of laparoscopic colorectal surgical studies.[6] The review identified three distinct categories of QA measures: (i) trial entry criteria for surgeons and centres, (ii) standardisation of surgical techniques and (iii) monitoring of surgeons and/or units. Despite this, the use of such QA measures is rarely reported. In addition, it did not consider how QA processes may differ between pragmatic and explanatory trials. A recent systematic review of 80 RCTs found that 18% used entry criteria for surgeons or centres, 29% attempted to standardise the surgical procedures under evaluation (although most did not describe what the standards were), and 28% undertook some form of monitoring during the trial.[7] An additional problem is that the QA processes were often selected arbitrarily and relied on surgeons' self-reported data, rather than objective measurements, which leaves

them open to criticism. Practical, robust approaches to QA in pragmatic surgical studies are lacking. The aim of this study, therefore, was to develop and test QA processes for pragmatic surgical trials, in the context of a predominantly pragmatic RCT evaluating surgical techniques in upper gastrointestinal cancer surgery (Randomised Oesophagectomy–Minimally Invasive or Open (ROMIO), HTA 14/140/78).[8 9]

### The ROMIO study
The purpose of the ROMIO study is to compare, in patients with cancer of the oesophagus and oesophago-gastric junction, the clinical and cost effectiveness of laparoscopically assisted (LAO) and open (OO) surgical procedures in terms of recovery, health-related quality of life, cost and survival. The RCT will be conducted in at least eight UK centres. The ROMIO study is predominantly pragmatic, as demonstrated by the Pragmatic-Explanatory Continuum Indicator Summary II wheel provided in figure 1.[10] For example, it is multicentre and involves more than 40 surgeons. It has broad inclusion criteria and it is expected that at least 60% of patients undergoing oesophagectomy will be eligible to participate. The primary outcome is patient centred and secondary outcomes include resource use and other clinical and patient-reported outcomes. Despite this, however, assessing QA is crucial, to ensure that the LAO is performed to a similar standard as the OO, enabling a fair comparison to be made between the two techniques. Within the ROMIO pilot study, work was undertaken to begin the process of establishing QA methods. This protocol outlines plans to test and assess the feasibility of implementing these QA methods for the purposes of a multicentre RCT in surgery.

### OBJECTIVES
► To examine the variability of performance of oesophagectomy and agree on standardisation of surgery that are acceptable for a predominantly pragmatic multicentre trial.
► To pilot a tool to assess the quality of oesophagectomy undertaken within the ROMIO study.
► To pilot a tool to enable ongoing monitoring of surgeons' technical performance throughout the RCT.
► To explore the feasibility of using intraoperative digital photography and videos as methods for assessing QA in an RCT.
► To develop a feedback system for surgeons participating in the trial.

### METHODS AND ANALYSIS
Methods to assess QA will be developed from work initiated in the pilot RCT. There are three categories: (i) entry criteria for surgeons and centres, (ii) standardisation of surgical techniques and (iii) monitoring of surgeons and centres during the trial. Methods to assess QA in each

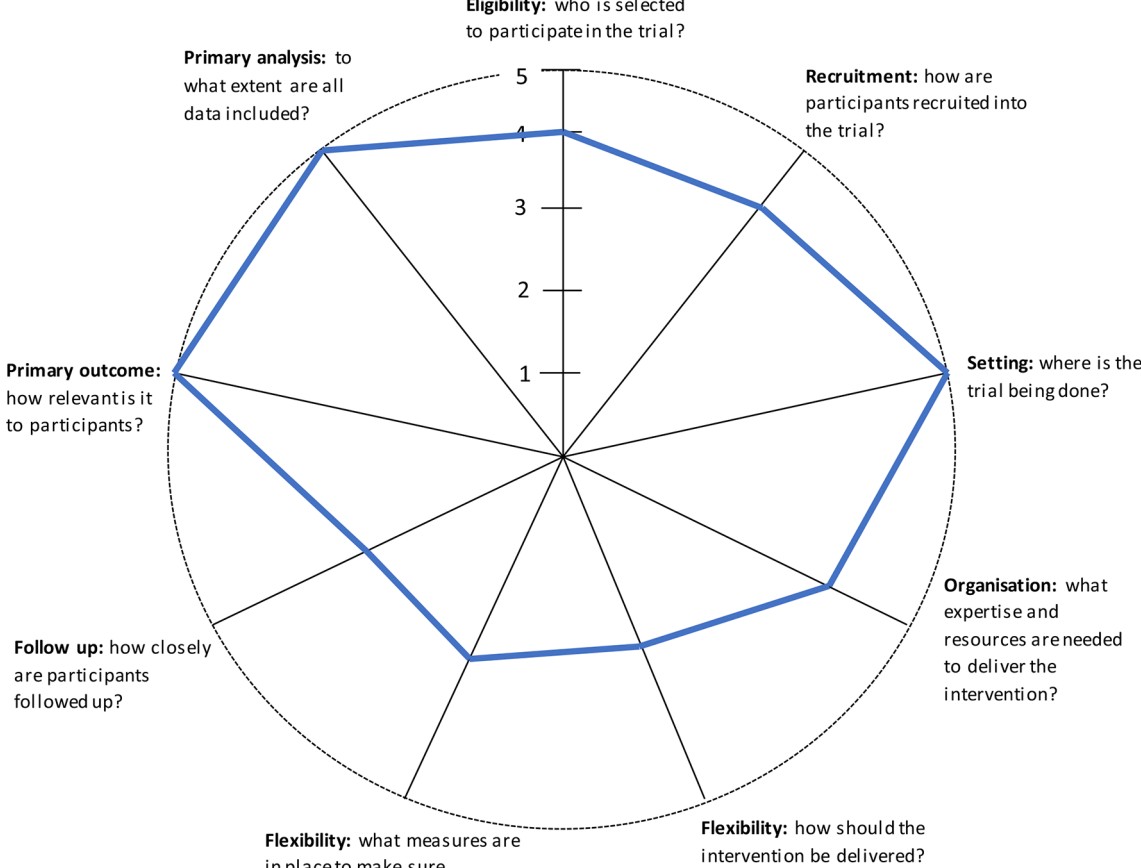

**Figure 1** Pragmatic-Explanatory Continuum Indicator Summary II wheel: a visual representation of the predominantly pragmatic nature of the Randomised Oesophagectomy–Minimally Invasive or Open study.[10] Trials that take an explanatory approach produce wheels nearer the hub; those with a pragmatic approach are closer to the rim.

of these categories will carefully consider the balance between extensive selection (of centres and surgeons) and standardisation (of technique) with the predominantly pragmatic nature of the study.

## ENTRY CRITERIA FOR CENTRES

The criteria used to select centres to participate in the ROMIO study will be based on discussions within the study management group, informed by: recommendations for cancer centres (>50 cases per year), experience of team working in trials (agreement of at least two surgeons to enter patients into the trial) and commitment (shown by the provision of centre-level data for submission to the National Oesophago-Gastric Audit).

## ENTRY CRITERIA FOR SURGEONS

Rather than prevent surgeons from participating in the study, the purpose of this aspect of QA was predominantly to enable (i) variations in surgical technique and skill to be described, facilitating contextualisation of the results, (ii) provision of feedback and (iii) to establish whether LAO was broadly being performed to the same standard as OO. The feasibility of collecting videos of the abdominal phase of OO will be established. If found

to be possible, participating surgeons will be required to submit one unedited video of the abdominal phase of OO (the 'standard' technique), in line with existing literature.[2] Because LAO represents a 'new' technique, and to prevent surgeons from selecting only their 'best' example, two unedited videos of the abdominal phase of LAO will be required. All videos will be pseudonymised.

A schema outlining the proposed development and validation of the video assessment tool is provided in figure 2. Current available methods for assessing the quality of surgeons' technical skills from videos include hierarchical task analysis[9] and the Objective Structured Assessment of Technical Skills (OSATS) tool.[11] OSATS is suitable for use with any type of surgical procedure (although has not been formally tested in the context of oesophagectomy) whereas hierarchical task analyses are developed individually for specific procedures. A hierarchical task analysis for oesophagectomy (HTA-O) was developed during the pilot phase of the ROMIO study[9]; however, it has not yet been formally tested. Both OSATS and HTA-O measures will be piloted simultaneously using 'think aloud' techniques, whereby a researcher will observe a surgeon (from the review team, see the 'Data analysis' section below) while reviewing an operative video.[12] The surgeon will be asked to complete each measure and express their

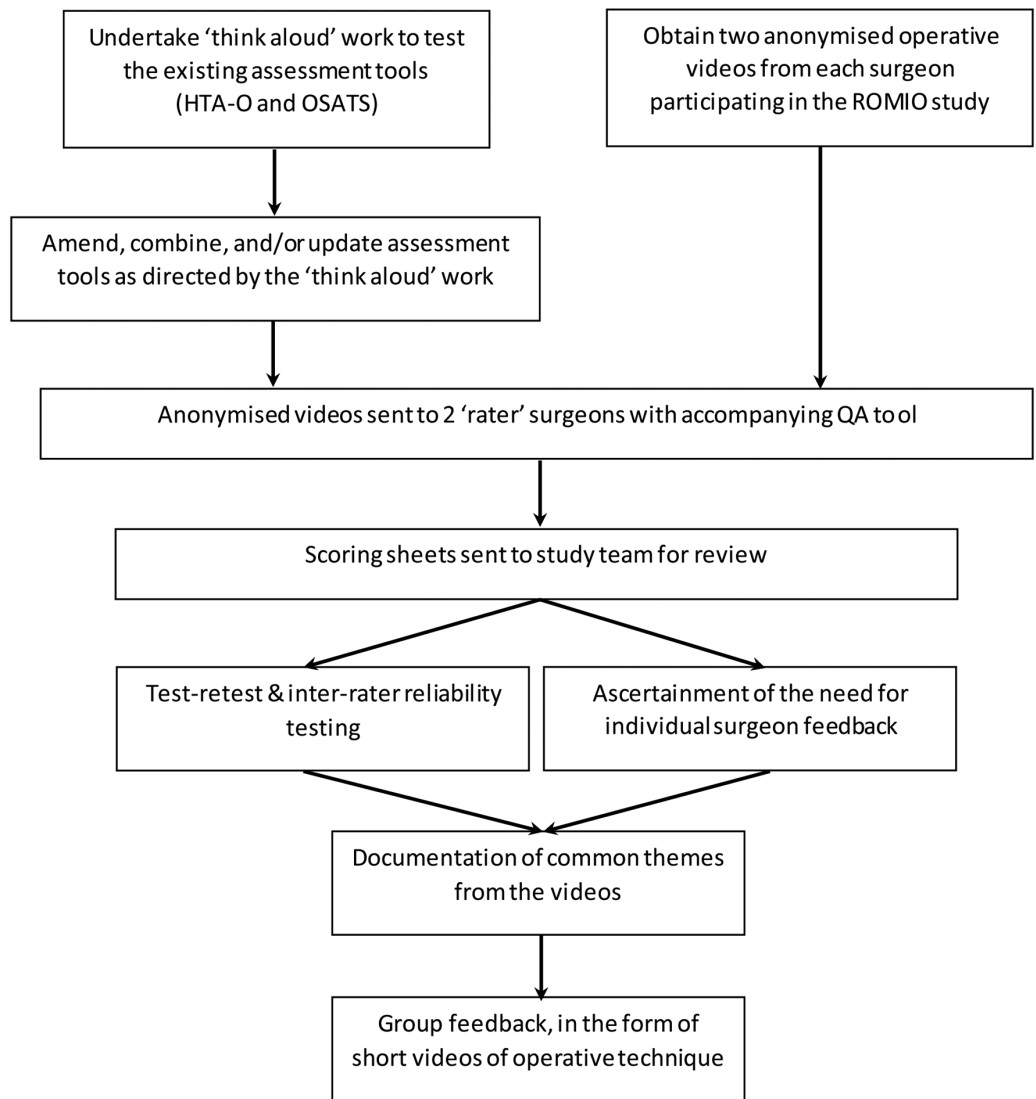

**Figure 2** Schema for the development of a pretrial QA tool to assess surgeons' overall operative technique in relation to oesophagectomy. HTA-O, Hierarchical Task Analysis for Oesophagectomy [8]; OSATS, Objective Structured Assessment of Technical Skills; QA, quality assurance; ROMIO, Randomised Oesophagectomy–Minimally Invasive or Open.[9]

thoughts while doing so, and to vocalise any general feelings about the technical skills displayed on the video. The 'think aloud' sessions will be audio recorded, transcribed verbatim and analysed thematically. Based on the findings from this process, the existing measures (OSATS and HTA-O) may be amended or combined, and new domains added. The 'think aloud' process will be iterative—that is, it will be repeated with different videos until the surgeon has no further comments—and stop once the study team are satisfied that no new amendments are necessary. It is anticipated that two surgeons will be involved in this piloting phase.

### STANDARDISATION OF SURGICAL TECHNIQUES

This phase of QA will be undertaken with careful consideration of the need to balance extensive standardisation with the pragmatic nature of the study, and the practical challenges associated with monitoring adherence to the

standards. LAO and OO will be deconstructed into their component parts, the 'key operative components' identified, and the expected similarities and differences for each component of both procedures will be mapped and documented using a typology of surgical interventions.[13] Details of how each component is recommended to be performed (and the degree of flexibility permitted) will be agreed based on evidence from existing literature and consensus among the study team.

### MONITORING OF SURGERY DURING THE TRIAL

A schema outlining the piloting of a 'photo metric' of intraoperative QA for oesophagectomy is provided in figure 3. The photo metric is based on the premise that identification of specific anatomical structures on a photograph can be used as a measure of the quality of operative technique (because if the quality is poor, the structures would not be clearly visible).[14] A list of anatomical

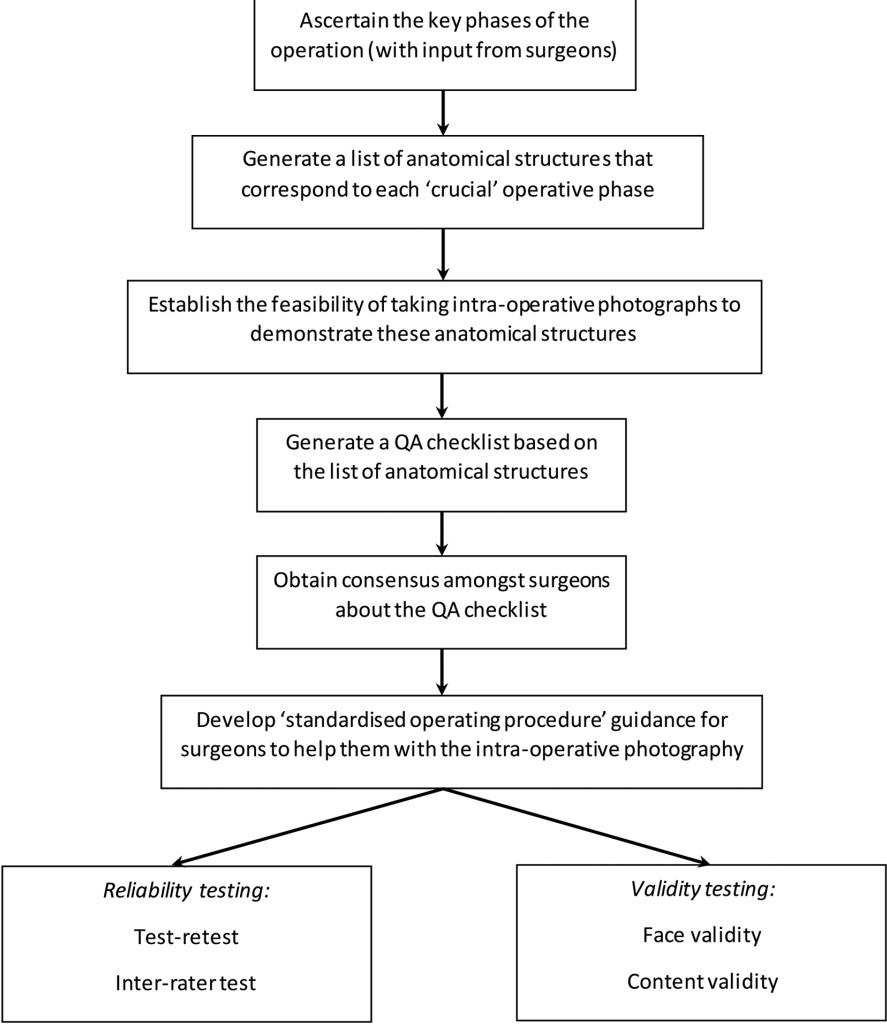

**Figure 3** Schema for the development of a QA tool (photo metric) to document the adequacy of crucial intraoperative steps building on work done in the pilot randomised controlled trial. QA, quality assurance.

structures that would be expected to be visible during each key component (as identified above) of LAO and OO was compiled during the pilot phase of the ROMIO study.[9] This provisional list will be refined by pilot testing in the operating theatre (out with the ROMIO study), to establish whether it is technically and logistically possible to collect photos of sufficient quality to demonstrate the required anatomical structures. The refined list will be discussed and agreed by surgeons participating in the ROMIO study. Subsequently, a rating scale will be developed (to include a category to represent that the anatomy cannot be seen or assessed) and the 'photo metric' will undergo reliability and validity testing. Surgeons will be encouraged to submit photos for each patient recruited to the ROMIO study, to demonstrate that the key operative components have been performed.

All trial pathology specimens will be prepared and macroscopically and microscopically assessed in a uniform manner. The pathology data for the trial will be collected using a standardised form and represent data points included within the Royal College of Pathologists Dataset. Data points that will serve as surgical QA indicators include the length of the oesophagus and the number of harvested lymph nodes. The slides of 10% of all cases from each centre will be reviewed by the lead pathologist. Pathologists will be blinded to the randomised allocation for each sample.

## DATA COLLECTION

Patients will be asked to give written informed consent for video and photographic recordings of the procedure (separate to consent for recruitment into the ROMIO study itself), and transfer of the data to the trial coordinating centre (using local and study consent forms). LAO videos will be collected directly from the laparoscopy 'stack' already in routine use for the operations. Video recording will start from when the surgeon inserts the camera port and will end when the camera is removed after the procedure. Processes for obtaining videos of OO will be established as part of this study. Photographs will be taken using the laparoscopy 'stack' (for LAO and OO) and collected for each of the key phases identified during the 'standardisation of surgical procedures' process (described above).

## Feasibility data

An additional objective of this study is to collect information about the feasibility of developing QA processes involving the collection of digital images. Technical issues (and their solutions, if appropriate) will be documented as they arise and 'standard operating procedure' documents will be produced for both the video and photography QA aspects of this study. We envisage that it may be particularly problematic to collect digital videos of open surgery, that sufficiently match the quality of laparoscopic procedures. Specific attention will be paid to developing solutions to overcome the barriers to collecting such data, including the need to minimise any interference with the usual process of surgery.

## Contextual information

We will record the total number of procedures performed by each participating surgeon, annual procedural volume and the total volume of oesophagectomies performed in each centre. The number of trainees undertaking procedures under supervision, and the number of procedures involving trainees, will be documented.

## DATA ANALYSIS

Rating of videos and photos will be undertaken by a team of oesophagogastric surgeons: two participating in the ROMIO study and three routinely undertaking oesophagectomy outside of the trial. Each video and photo will be rated by at least two surgeons. As with similar previous studies,[2] no formal training or guidance will be given regarding the assessment tools. The 'rater' surgeons will not be aware of the surgeon or centre from which the videos and photos were obtained.

### Video assessments (surgeon entry criteria)

Where videos are incomplete (eg, a component of the operation has not been captured), the surgeon will be asked to provide a further video. Videos of surgeons receiving a summary judgement of 'poor' skill (or worse) will automatically be discussed and reviewed by the study team.

### Photo assessments (monitoring)

The scoring system will be used to establish a threshold at which the standard of surgery is considered 'sufficient'. This will be determined by iterative review of all photos in discussion with members of the study management group. All photos considered not to meet this standard will be reviewed by the study team. Photo assessments will be correlated with the corresponding lymph node yield for each patient.

## FEEDBACK
### Individual feedback

In cases where videos or photos do not meet the expected quality (in terms of the quality of the images as well as the standard of surgery), individualised and private feedback will be provided. In such instances, further operative videos may be requested to clarify that the feedback points have been addressed. However, given that the participants are all consultant oesophagogastric surgeons, we anticipate that this will rarely occur.

## Group feedback

To improve the overall quality of operations, which is another important part of QA, we will develop generic feedback materials for participating surgeons and centres. The generic feedback materials will be developed in two phases: (i) understanding operative techniques, variations and difficulties (by watching the videos and documenting emerging patterns and themes) and (ii) developing an action plan to address the identified difficulties and optimise operative techniques. We will achieve the second phase by developing short videos demonstrating operative techniques, based on the themes detected across the surgeons' submitted videos. These videos will be sent to all participating surgeons at regular intervals during the study. The key issues will also be discussed at ROMIO study investigators' meetings where exemplar videos of 'excellence' and 'room for improvement' will be displayed to allow self-reflection and learning.

After participating in the ROMIO study for 12 months, we will ask surgeons to submit further operative videos, to review progress and standards.

## FUTURE WORK

This study aims to develop methods for measuring the QA of surgical interventions. Once developed, they will be implemented in the context of an RCT comparing open and laparoscopic surgery for patients with oesophageal cancer. There are numerous analyses that may be undertaken, which will depend on the exact nature of the QA measures that are developed. First, the quality of intervention delivery will be assessed, and to compare what surgeons reported in the case report forms (ie, what they said they did) with the intraoperative photographs (ie, what actually happened). Second, we will explore trends relating to surgeon skill (from the operative videos) and patient outcomes, though numbers are relatively small and this may not, therefore, be possible. Third, the process of developing QA measures for this study will influence future work in this area; specifically, the generation of guidance that can be extrapolated to other RCTs in surgery. This is the focus of a funded fellowship award, which will also examine the process of obtaining consensus about exactly what these QA measures should comprise, accounting for trial design and the nature of the interventions under investigation. Finally, it is important to recognise that improving the QA of surgical interventions may improve standards of surgery within an RCT. While we will document the points at which feedback is given to surgeons and centres, it may not be realistically possible to correlate this (and assign causation) with outcomes. It is well recognised that patients within RCTs generally have more favourable outcomes than those that do not. Improvements to the standards of intervention delivery

may, therefore, form a part of the benefits associated with trial participation.

## Public and patient involvement

Patients and the public were extensively involved in the design of the ROMIO study. During these meetings, we asked for their views about the QA aspect. They felt it was an important aspect of the RCT and did not have any issues relating to the acquisition of video and photo data relating to their operation.

## ETHICS AND DISSEMINATION

Digital videos will be transferred using Open Document Information Exchange (ODIE) to the National Health Service (NHS) network for analysis by the ROMIO study team and pseudonymised with a unique identifier. The ODIE file hosting system is securely protected through use of an encrypted HTTPS link, meaning third parties cannot read exchanged data. Digital photographs will be uploaded directly by the local site staff into the purpose-designed server ROMIO database hosted on the NHS network. Information capable of identifying individuals will be held in the database with passwords restricted to ROMIO study staff.

Results of the study will be submitted for publication in a peer-reviewed journal and presented at national and international gastrointestinal conferences. Guidance documents relating to: (i) practical considerations for the development of QA procedures in surgical RCTs and (ii) generation of feedback materials to improve QA, will also be published. Feedback materials (in the form of 'gold standard' intraoperative photos of each key phase and operative technique videos) will also be made available.

### Author affiliations
[1]Centre for Surgical Research, School of Social and Community Medicine, University of Bristol, Bristol, UK
[2]Division of Surgery, University Hospitals Bristol NHS Foundation Trust, Bristol, UK
[3]Medical Illustration, University Hospitals Bristol NHS Foundation Trust, Bristol, UK
[4]Bristol Randomised Trials Collaboration & School of Social and Community Medicine, University of Bristol Faculty of Medicine and Dentistry, Bristol, Bristol, UK
[5]Clinical Trials and Evaluation Unit, University of Bristol, Bristol, UK
[6]Department of Upper GI Surgery, Gloucestershire Hospitals NHS Foundation Trust, Gloucester, UK
[7]Department of Upper GI Surgery, Beaumont Hospital, Dublin, Ireland
[8]Department of Upper GI Surgery, University Hospitals of Leicester NHS Trust, Leicester, UK
[9]Faculty of Medicine, Department of Surgery & Cancer, Imperial College, London, UK
[10]Division of Surgery, University Hospitals Bristol NHS Foundation Trust, Bristol, UK

**Collaborators** The ROMIO Study team is comprised of the following groups: Co-applicants: Chris Metcalfe, Jenny Donovan, Newton Wong, Richard Berrisford, Benjamin Howes, William Hollingworth, Chris Rogers, Kerry Avery, Jackie Elliott. Other members of the ROMIO study: Lucy Culliford, Marcus Jepson, Peter Lamb, Ravinder Vohra, JamesCatton, Rachel Melhado, Kish Pursani, Richard Krysztopik, James Byrne, Bilal Alkhaffaf,Tim Underwood, PaulWilkerson , Christopher Streets , Dan Titcomb , Richard Berrisford , Lee Humphreys , Tim Wheatley , Grant Sanders , Arun Ariyarathenam , James Byrne, Jamie Kelly, FergusNoble, Graeme Couper, Richard Skipworth , Chris Deans, Anna Paisley, Sukhir Ubhi,Rob Williams, David Bowrey, David Exon, Paul Turner, Vinutha Shetty, RamChaparala, Khurshid Akhtar, Siba Senapati, Simon Parsons, Neil Welch, NaheedFarooq.

**Contributors** All authors contributed to the study design and approved the final manuscript. NSB, JB, AS, DG, RB, GH, CPB: developed the video and photo tools. AH, SD, SH, WR, AB, CPB: tested the video tools and have undertaken assessments. NB, JB: wrote the first draft of the manuscript.

**Funding** The ROMIO study (Randomised Oesophagectomy–Minimally Invasive or Open) is funded by the NIHR HTA (HTA 14/140/78). The quality assurance aspect of the ROMIO study was supported by the MRC ConDuCT-II (Collaboration and innovation for Difficult and Complex randomised controlled Trials In Invasive procedures) Hub for Trials Methodology Research (MR/K025643/1), and the NIHR Biomedical Research Centre at the University Hospitals Bristol NHS Foundation Trust and the University of Bristol.

**Disclaimer** The views expressed in this publication are those of the author(s) and not necessarily those of the NHS, the National Institute for Health Research or the Department of Health. JMB is an NIHR Senior Investigator and NSB is an NIHR Clinical Lecturer.

**Competing interests** None declared.

**Patient consent for publication** Not required.

**Ethics approval** Ethical approval for the quality assurance work was obtained as part of the ROMIO study.

**Provenance and peer review** Not commissioned; externally peer reviewed.

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
