## [Reviewer comments · BMJ Open]

ARTICLE DETAILS

TITLE (PROVISIONAL)	A protocol for developing quality assurance measures to use in surgical trials: an example from the ROMIO study
AUTHORS	Blencowe, Natalie; Skilton, Anni; Gaunt, Daisy; Brierley, Rachel; Hollowood, Andrew; Dwerryhouse, Simon; Higgs, Simon; Robb, William; Boddy, Alex; Hanna, George; Barham, C; Blazeby, Jane

VERSION 1 – REVIEW

REVIEWER	Lawrence Kim University of North Carolina, USA
REVIEW RETURNED	04-Oct-2018

GENERAL COMMENTS	This manuscript describes methods to be undertaken for quality control in a surgical trial comparing two different operative procedures. The authors are to be commended for thinking critically about quality control in this challenging study. Because it is a methods paper, there are no results about which to comment. The process for developing quality control measures is described, but the criteria for success are unclear. The work described is standard in clinical trials involving surgical procedures. Usually the process is described after it is completed. I am not sure that a description of the process before it is done is useful to the reader. I have two major concerns about the process described. First is the conflict between standardization and generalizability. The authors do acknowledge this conflict. However, the extensive efforts that will be undertaken for quality control and standardization lead me to conclude that this is not, in fact, a pragmatic trial. I am not arguing either for or against standardization as there are advantages each way. But I would not call the end result a pragmatic trial. Second, the extensive observation is quite intrusive into the normal conduct of surgery. Video capture of open surgical procedures is notoriously difficult, and the quality of video from the laparoscopic recording and the open case videos is likely to be vastly different. This introduces one form of bias. Furthermore, the highly intensive supervision will maximize the Hawthorne effect, thereby reducing the generalizability of the trial. Surgeons will be ever aware that their every move is being watched and dissected on video. This almost ensures that the surgeons will be altering their behavior in innumerable small and poorly defined ways. This may meaningfully influence the outcome of the procedures, either positively or negatively. This again is contrary to what I usually understand as a pragmatic trial.
--

	I would suggest that the authors either abandon the notion that this is a pragmatic trial, or significantly reduce their efforts at quality control and standardization. You can't have it both ways.
--	---

REVIEWER	Paco Welsing UMC Utrecht, the Netherlands
REVIEW RETURNED	17-Oct-2018

GENERAL COMMENTS	The authors discuss a highly relevant issue of the practice of performing Pragmatic trials and quality of the intervention in surgery trials. Methodology used to capture the 'quality' of surgery seems adequate. - An important issue here which, I think, should be better described in the manuscript is the inherent contradiction between the pragmatic nature of the trial, where the aim is to provide results directly generalizable to daily practice' and the concept of obtaining quality assurance information, actions taken based on this information, and selection of participating surgeons based on performance. For instance how should this be discussed in presenting results of the trial and guidelines/recommendations based on trial results? - It is not clear to what level entry criteria for centers and for surgeons are actually used in the execution of the ROMIO study, or whether only measures to base a (future) selection strategy on are collected and (in a separate study/analysis/evaluation) are used to evaluate strategies for selection for (future) trials. E.g. on p6 it is simply stated that that results will be used to 'inform the debate around standards of surgery in the trial' If this is done in the trial it will need to be done (long) before the trial can be started (as it first needs to be clear whether the surgeon satisfies the entry criteria). Specific comments - p6line 22: 'surgeon' should be 'researcher? -p6, line 33: the 'think aloud' process will be iterative... ' Does this mean that different researchers/surgeons (the two participating?) will repeatedly look at the video material until no different remarks will be made by them? -p7, line 44/45: indeed the balance between extensive standardization with the pragmatic nature of the study is key, however, I think this is also the case for the selection of centers and surgeons.. - p8, line 26: Does pilot testing occur in the trial, or is this in the ROMIO trial. line 34: Is the 'photo metric' the QA checklist in Figure 2? Consider harmonize wording. -p9 first line: Is the informed consent regarding photographic recordings part of the inclusion criteria for the ROMIO study or is it optional for the study on piloting the quality assurance methods in this trial? - Analysis mainly seem qualitatively, based on discussions. Are there any more quantitative analyses planned, i.e regarding the relations between quality metrics/subgroups and trial outcome? - p10/11, Feedback during the trial can influence results in the trial, how should this be handled in the analysis and interpretation of results of such a trial?
--

REVIEWER	Elaine Irving
-----------------	---------------

	GlaxoSmithKline R&D Ltd, UK
REVIEW RETURNED	24-Oct-2018

GENERAL COMMENTS	The objective of the study is to develop QA measures that can be used in pragmatic clinical trials, however there are a number of aspects that that the authors should address/consider 1) The objective of the study is to create QA measures for use in a pragmatic trial. The aim of a pragmatic trial is to compare techniques in a real world setting and therefore variation in surgical approaches/techniques is part of this real word environment. The approach being taken here will intervene and aim to standardise these surgical techniques and approaches and therefore potentially impact the outcome of the pragmatic trial i.e. feedback on performance to consultant surgeons you will improve the standard of surgery and therefore impact the real world setting where this feedback does not routinely happen. Perhaps the approach should be more about describing variability rather than intending to increase quality in these trials 2) The protocol describes the use of photos and video footage as QA measures however more justification should be given regarding the sampling techniques proposed to demonstrate that the samples taken provide representative evidence of performance. 3) The protocol should describe how the data from the QA measures will be analysed and used to develop and validate a set of QA measures for future pragmatic trials and also describe how agreement will be reached on the use of such measures in future pragmatic trials.
---

VERSION 1 – AUTHOR RESPONSE

Reviewer 1 comments

1. This manuscript describes methods to be undertaken for quality control in a surgical trial comparing two different operative procedures. The authors are to be commended for thinking critically about quality control in this challenging study.

Reply: We thank the reviewer for their support.

2. Because it is a methods paper, there are no results about which to comment. The process for developing quality control measures is described, but the criteria for success are unclear.

Reply: We apologise that the reviewer found the criteria for success unclear. We have outlined the aims of the study and provided a list of objectives (page 5). We have now modified the aim of the study to clarify this.

Revisions: The aim of the study (page 4) now reads:

“The aim of this study, therefore, was to develop and test the feasibility of implementing QA processes for pragmatic-surgical trials, in the context of an RCT evaluating surgical techniques in upper GI cancer surgery (ROMIO: Randomised Oesophagectomy – Minimally Invasive or Open, HTA 14/140/78)⁷.”

~~“This protocol outlines plans to test and assess the feasibility of implementing and modify these QA methods for the purposes of a multicentre, pragmatic RCT in surgery the findings from which are required to be as generalisable as possible.”~~

3. The work described is standard in clinical trials involving surgical procedures. Usually the process is described after it is completed. I am not sure that a description of the process before it is done is useful to the reader.

Reply: We thank the reviewer for this comment and will attempt to persuade them that there are numerous benefits in publishing a methods paper. Quality assurance in surgical RCTs is an extremely topical issue. One of the main criticisms levelled at surgical RCTs relates to the quality of intervention delivery, including surgeon expertise. This paper describes three key ways to improve the quality and transparency of intervention delivery: trial entry criteria (for surgeons and centres), intervention description and standardisation, and monitoring of intervention delivery during the trial. Crucially, these QA measures are all based on evidence rather than many current measures. We believe these measures are transferrable to other RCTs evaluating surgical interventions. Publishing the protocol for the methods in detail as a separate paper will mean that the paper showing the results will refer to this and thus have more space to report new data.

4. I have two major concerns about the process described. First is the conflict between standardization and generalizability. The authors do acknowledge this conflict. However, the extensive efforts that will be undertaken for quality control and standardization lead me to conclude that this is not, in fact, a pragmatic trial. I am not arguing either for or against standardization as there are advantages each way. But I would not call the end result a pragmatic trial.

Reply: We thank the reviewer for raising this important issue and acknowledge the difficulties relating to standardization and quality assurance more generally in pragmatic trials. The problem with a lack of quality assurance relating to the surgical intervention is that it can be impossible to know how they were delivered in a trial, introducing performance bias and compromising internal validity. We believe that this is a great threat to a trial's results and is currently one of the main reasons why surgical trials are criticised. We acknowledge that 'too much standardization' may be detrimental to the generalisability of a pragmatic trial and that this needs to be balanced with the need to maximise rigour and internal validity. Moreover, it is rare for a study to be purely pragmatic or explanatory. It is important to note that the methods described allow quality assurance to be tightly controlled or otherwise (depending on the agreed parameters for standardisation) in light of these issues we have made numerous changes to the manuscript. With regards to the ROMIO study specifically, we have 'toned down' the assertion that it is a purely pragmatic study.

Revisions: Within the abstract (page 3), the objective now reads:

Objective: This protocol describes a process to develop and test quality assurance (QA) measures for use within a predominantly pragmatic surgical RCT comparing minimally invasive and open techniques for oesophageal cancer (the NIHR ROMIO study).

Throughout the manuscript, we have changed references to the ROMIO study from 'pragmatic' to 'predominantly pragmatic'

The introduction section has been extensively re-worked. A new paragraph has been added to provide further detail about the need for quality assurance of surgical interventions:

“Randomized controlled trials (RCTs) in surgery are notoriously difficult to design and conduct, due to numerous methodological and cultural challenges. Many of these challenges relate to the fact that surgical procedures are complex healthcare interventions, meaning that ‘unlike 20 milligram tablets, no

two procedures are the same' and achieving standardization of surgical techniques and processes is difficult¹. This is partly because surgeons naturally undertake procedures in (slightly) different ways and have differing skill levels, which ~~There is emerging empirical evidence that surgeons' technical skill~~ may influence rates of post-operative complications and re-operation². A lack of consideration for intervention standardization and surgeon expertise in the context of RCTs may introduce bias, compromising internal validity. This is acknowledged in guidance such as CONSORT-NPT (an extension to the CONSORT statement for non-pharmacological treatments such as surgery) and SPIRIT, which provides a checklist of 33 items to be reported in trial protocols. They recommend reporting precise details of the intervention and its components to enable replication in routine practice, information about standardization, procedures for monitoring adherence to intervention protocols, and consideration for the expertise of care providers.

Neither CONSORT-NPT nor SPIRIT differentiate between the information required in pragmatic or explanatory settings. This may be because it is rare for trials to be purely pragmatic or explanatory: trial design is not dichotomous and there is a continuum between the two extremes. In explanatory trials, which determine the efficacy of interventions, great detail may be necessary because the interventions are often novel and their safety needs to be assessed within carefully controlled settings. Pragmatic trials, which determine whether interventions are effective in the real world, are often multicentre studies with large numbers of surgeons³. Under such circumstances, specifying each operative step is likely to create difficulties, and ensuring that each step was delivered as planned may be unrealistic. A balance between adequate standardization and practicality is therefore necessary and appropriate. One way of achieving this is to determine the minimum active ingredients of the intervention – those that are thought to optimize outcomes or those that are different between the interventions in each trial group – and the degree to which they need to be standardized. In this way, monitoring only the key components may be sufficient, rather than monitoring all components and steps, in order to ensure the intervention is actually delivered as planned. This approach would also account for the fact that most trials sit within the pragmatic-explanatory continuum rather than being one or the other.

It is, therefore, important to provide reassurance about the standards of surgery in all RCTs, whilst recognising that this may vary according to trial design. One way of achieving this is to undertake quality assurance, defined as the process(es) of 'directing the performance and behaviours of practitioners and institutions toward more appropriate and acceptable health outcomes'⁴....."

The section about the ROMIO study has been expanded to provide more information about why the study is 'predominantly pragmatic':

"The purpose of the ROMIO study is to compare, in patients with cancer of the oesophagus and oesophago-gastric junction, the clinical and cost-effectiveness of laparoscopically assisted (LAO) and open (OO) surgical procedures in terms of recovery, health related quality of life, cost and survival. The RCT will be conducted in at least eight UK centres. The ROMIO study is predominantly pragmatic, as demonstrated in Figure 1. For example, it is multi-centre and involves more than 40 surgeons. It has broad inclusion criteria and it is expected that at least 60% of patients undergoing oesophagectomy will be eligible to participate. The primary outcome is patient-centred and secondary outcomes include resource use and other clinical and patient reported outcomes. Despite this, however, assessing quality assurance (QA) is crucial, to ensure that the laparoscopically-assisted procedure (LAO) is performed to a similar standard as the open operation (OO), enabling a fair comparison to be made between the two techniques."

A new figure (Figure 1) has been added, to provide a visual depiction of the predominantly pragmatic nature of the ROMIO study. This was compiled using PRECIS-II (PRagmatic Explanatory Continuum Indicator Summary) which is a tool designed to help trialists to consider where their trial is on the pragmatic/explanatory continuum.

Figure 1. PRECIS-II wheel: a visual representation of the predominantly pragmatic nature of the ROMIO study

5. Second, the extensive observation is quite intrusive into the normal conduct of surgery. Video capture of open surgical procedures is notoriously difficult, and the quality of video from the laparoscopic recording and the open case videos is likely to be vastly different. This introduces one form of bias. Furthermore, the highly intensive supervision will maximize the Hawthorne effect, thereby reducing the generalizability of the trial. Surgeons will be ever aware that their every move is being watched and dissected on video. This almost ensures that the surgeons will be altering their behaviour in innumerable small and poorly defined ways. This may meaningfully influence the outcome of the procedures, either positively or negatively. This again is contrary to what I usually understand as a pragmatic trial.

Reply: We agree with the reviewer that obtaining digital videos of open procedures is likely to be much more challenging than for laparoscopic surgery. For this reason, our study aims to assess the feasibility of capturing open oesophagectomy whilst recognising that we may conclude that it is not possible. We would like to reassure the reviewer that the Hawthorne effect is likely to be minimal in the context of the operating theatre. One of the authors undertook similar work during her PhD and found that surgeons tend to forget they are being videoed after only a few minutes (Blencowe *et al* Trials 2015; 16: 589). Video recording and photographing operations is becoming increasingly common; for example, the Netherlands and the United States routinely collect such data in gallbladder and bariatric surgery, respectively. Many operating theatres in the UK already have cameras in the overhead theatre lights. In addition, the constant advances in digital imaging and data capture mean that it is likely that these approaches will be used much for widely in training and in clinical practice. For these reasons, we believe that this feasibility work is worth pursuing.

Revisions: We have added a bullet point into the 'strength and limitations of the study' section:

"This study will explore the feasibility of obtaining digital videos of open surgery during this study; however, it may not be possible to achieve this with good enough quality or without interfering with the process of surgery"

We have emphasised the feasibility aspect of collecting open surgery videos in the 'feasibility data' section, page 10:

"An additional objective of this study is to collect information about the feasibility of developing QA processes involving the collection of digital images. Technical issues (and their solutions, if appropriate) will be documented as they arise and 'standard operating procedure' documents will be produced for both the video and photography QA aspects of this study. We envisage that it may be particularly problematic to collect digital videos of open surgery, that sufficiently match the quality of laparoscopic procedures. Specific attention will be paid to developing solutions to overcome the barriers to collecting such data, including the need to minimise any interference with the usual process of surgery."

6. I would suggest that the authors either abandon the notion that this is a pragmatic trial, or significantly reduce their efforts at quality control and standardization. You can't have it both ways.

Reply: Again we thank the author for this insight about the conflict between standardization and pragmatic trials. We have considered this carefully and made extensive revisions to the manuscript in light of these comments.

Revisions: Please refer to comment 5 for details of the revisions we have made regarding this point.

Reviewer 2 comments

1. The authors discuss a highly relevant issue of the practice of performing pragmatic trials and quality of the intervention in surgery trials. Methodology used to capture the 'quality' of surgery seems adequate.

Reply: We thank the reviewer for their support.

2. An important issue here which, I think, should be better described in the manuscript is the inherent contradiction between the pragmatic nature of the trial, where the aim is to provide results directly generalizable to daily practice and the concept of obtaining quality assurance information, actions taken based on this information, and selection of participating surgeons based on performance. For instance how should this be discussed in presenting results of the trial and guidelines/recommendations based on trial results?

Reply: We thank the reviewer for this comment and have considered it carefully, especially since it was raised by all three reviewers. We agree that we did not discuss the issue of standardisation and generalizability in enough detail.

Revisions: We have expanded the introduction to include information about explanatory and pragmatic trials, and the tensions between quality assurance and generalisability. Please refer to our reply to point 5 of reviewer 1's comments for a full explanation.

3. It is not clear to what level entry criteria for centres and for surgeons are actually used in the execution of the ROMIO study, or whether only measures to base a (future) selection strategy on are collected and (in a separate study/analysis/evaluation) are used to evaluate strategies for selection for (future) trials. E.g. on p6 it is simply stated that that results will be used to 'inform the debate around standards of surgery in the trial' If this is done in the trial it will need to be done (long) before the trial can be started (as it first needs to be clear whether the surgeon satisfies the entry criteria).

Reply: We apologise for this lack of clarity and have now re-worded this paragraph accordingly.

Revisions: The paragraph 'entry criteria for surgeons' now reads:

"Rather than prevent surgeons from participating in the study, the purpose of this aspect of QA was predominantly to enable i) variations in surgical technique and skill to be described, facilitating contextualisation of the results, ii) provision of feedback, and iii) to establish whether LAO was broadly being performed to the same standard as OO. The feasibility of collecting videos of the abdominal phase of OO will be established. If found to be possible, participating surgeons will be required to submit one unedited video of the abdominal phase of OO (the 'standard' technique), in line with existing literature². Because LAO represents a 'new' technique, and to prevent surgeons from selecting only their 'best' example, two unedited videos of the abdominal phase of LAO will be required. All videos will be pseudonymised."

4. p6, line 22: 'surgeon' should be 'researcher'?

Reply: We would like to clarify this; however, we cannot find the word 'surgeon' or 'researcher' on page 6, line 22. Would it be possible to provide us with the full sentence please?

5. p6, line 33: the 'think aloud' process will be iterative... ' Does this mean that different researchers/surgeons (the two participating?) will repeatedly looked at the video material until no different remarks will be made by them?

Reply: We thank the reviewer for this comment and would like to clarify that the 'think aloud' process will be repeated with the same two surgeons and different videos, until they have no further comments.

Revisions: The penultimate sentences of the 'entry criteria for surgeons' section, page 7, now read:

"The 'think aloud' process will be iterative – i.e. it will be repeated with different videos until the surgeon has no further comments - and stop once the study team are satisfied that no new amendments are necessary. It is anticipated that two surgeons will be involved in this piloting phase."

6. p7, line 44/45: indeed the balance between extensive standardization with the pragmatic nature of the study are key, however, I think this is also the case for the selection of centres and surgeons.

Reply: We thank the reviewer for this point and completely agree that this balance is also crucial for the selection of centres and surgeons. We have also clarified that the rationale for collecting operative videos in the 'entry criteria for surgeons' section is predominantly to describe the variations in technique and ability, rather than prevent surgeons from participating in the trial.

Revisions: We have added a sentence into the beginning of the Methods section to reflect this. The first paragraph of the Methods, page 6, now reads:

"Methods to assess QA will be developed from work initiated in the pilot RCT. There are three categories: i) entry criteria for surgeons and centres, ii) standardization of surgical techniques, and iii) monitoring of surgeons and centres during the trial. Methods to assess QA in each of these categories will carefully consider the balance between extensive selection (of centres and surgeons) and standardization (of technique) with the pragmatic nature of the study."

The final paragraph on page 6 - 'entry criteria for surgeons' – now reads:

"Rather than prevent surgeons from participating in the study, the purpose of this aspect of QA was predominantly to enable i) variations in surgical technique and skill to be described, facilitating the contextualisation of results, ii) provision of feedback, and iii) to establish whether LAO was broadly being performed to the same standard as OO. The feasibility of collecting videos of the abdominal phase of OO will be established. If found to be possible, participating surgeons will be required to submit one unedited video of the abdominal phase of OO (the 'standard' technique), in line with existing literature². Because LAO represents a 'new' technique, and to prevent surgeons from selecting only their 'best' example, two unedited videos of the abdominal phase of LAO will be required. All videos will be pseudonymised."

7. p8, line 26: Does pilot testing occur in the trial, or is this in the ROMIO trial?

Reply: We apologise for this lack of clarity.

Revisions: This sentence now reads:

“This provisional list will be refined by pilot testing in the operating theatre (out with the ROMIO study), to establish whether it is technically and logistically possible to collect photos of sufficient quality to demonstrate the required anatomical structures.”

8. line 34: Is the 'photo metric' the QA checklist in Figure 2? Consider harmonize wording.

Reply: We apologise for this discrepancy and have now aligned the terminology.

Revisions: The title of Figure 2 now reads:

“Figure 2. Schema for the development of a QA tool (photo metric) to document the adequacy of crucial intra-operative steps building on work done in the pilot RCT”

9. p9 first line: Is the informed consent regarding photographic recordings part of the inclusion criteria for the ROMIO study or is it optional for the study on piloting the quality assurance methods in this trial?

Reply: Informed consent for the photographic recordings is separate to recruitment into the ROMIO study, meaning that patients can choose not to have photographs taken during their operation but still participate in the trial.

Revisions: We have clarified this in the first sentence of the paragraph 'data collection, page 10:

“Patients will be asked to give written informed consent for video and photographic recordings of the procedure (separate to consent for recruitment into the ROMIO study itself), and transfer of the data to the trial co-ordinating centre (using local and study consent forms).”

10. Analysis mainly seem qualitative, based on discussions. Are there any more quantitative analyses planned, i.e regarding the relations between quality metrics/subgroups and trial outcome?

Reply: We thank the reviewer for raising this question. We have not yet written an analysis plan for the results arising from using the quality assurance measures in the ROMIO study. The analyses in this paper relate only to the development of the QA measures.

Revisions: We have added a section relating to analysis plans for the QA measures, entitled 'future work', page 12:

Future work

“This study aims to develop methods for measuring the QA of surgical interventions. Once developed, they will be implemented in the context of an RCT comparing open and laparoscopic surgery for patients with oesophageal cancer. There are numerous potential analyses that may be undertaken, which will depend on the exact nature of the QA measures that are developed. First, the quality of intervention delivery will be assessed, and to compare what surgeons reported in the case report forms (i.e. what they said they did) with the intra-operative photographs (i.e. what actually happened). Second, we will explore trends relating to surgeon skill (from the operative videos) and patient outcomes, though numbers are relatively small and this may not, therefore, be possible. Third, the process of developing QA measures for this study will influence future work in this area; specifically, the generation of guidance that can be extrapolated to other RCTs in surgery. This is the focus of a funded fellowship award, which will also examine the process of obtaining consensus about exactly what these QA measures should comprise, accounting for trial design and the nature of the interventions under investigation.”

11. p10/11, Feedback during the trial can influence results in the trial, how should this be handled in the analysis and interpretation of results of such a trial?

Reply: The reviewer raises an important point. Whilst we will document the points at which feedback is given, it may not realistically be possible to correlate this with outcomes. It is well recognised that patients participating in RCTs have more favourable outcomes than those out with. Improvements to the standards of intervention delivery may, therefore, form a part of this (and may in fact occur naturally over time as surgeons acquire additional expertise).

Revisions: We have added some information into a 'future work' section at the end of the manuscript to reflect this:

"Future work

This study aims to develop methods for measuring the QA of surgical interventions. Once developed, they will be implemented in the context of an RCT comparing open and laparoscopic surgery for patients with oesophageal cancer. There are numerous potential analyses that may be undertaken, which will depend on the exact nature of the QA measures that are developed. First, the quality of intervention delivery will be assessed, and to compare what surgeons reported in the case report forms (i.e. what they said they did) with the intra-operative photographs (i.e. what actually happened). Second, we will explore trends relating to surgeon skill (from the operative videos) and patient outcomes, though numbers are relatively small and this may not, therefore, be possible. Third, the process of developing QA measures for this study will influence future work in this area; specifically, the generation of guidance that can be extrapolated to other RCTs in surgery. This is the focus of a funded fellowship award, which will also examine the process of obtaining consensus about exactly what these QA measures should comprise, accounting for trial design and the nature of the interventions under investigation. Finally, it is important to recognise that improving the QA of surgical interventions may improve standards of surgery within an RCT. Whilst we will document the points at which feedback is given to surgeons and centres, it may not be realistically possible to correlate this (and assign causation) with outcomes. It is well recognised that patients within RCTs generally have more favourable outcomes than those that do not. Improvements to the standards of intervention delivery may, therefore, form a part of the benefits associated with trial participation."

Reviewer 3 comments

The objective of the study is to develop QA measures that can be used in pragmatic clinical trials, however there are a number of aspects that that the authors should address/consider:

1. The objective of the study is to create QA measures for use in a pragmatic trial. The aim of a pragmatic trial is to compare techniques in a real world setting and therefore variation in surgical approaches/techniques is part of this real world environment. The approach being taken here will intervene and aim to standardise these surgical techniques and approaches and therefore potentially impact the outcome of the pragmatic trial i.e. feedback on performance to consultant surgeons you will improve the standard of surgery and therefore impact the real world setting where this feedback does not routinely happen. Perhaps the approach should be more about describing variability rather than intending to increase quality in these trials

Reply: We completely agree with the reviewer and acknowledge that we did not discuss these issues in sufficient detail in the original version of our manuscript.

Revisions: In line with comments from reviewers 1 and 2, we have made extensive changes to reflect this feedback. Please refer to comment 5 from reviewer 1 for details of the changes we have made.

2. The protocol describes the use of photos and video footage as QA measures however more justification should be given regarding the sampling techniques proposed to demonstrate that the samples taken provide representative evidence of performance.

Reply: We thank the reviewer for this comment. We intend to collect intra-operative photos of the key surgical steps for *all* patients recruited to the ROMIO study. We apologise this was not clear.

Revisions: The following sentence has been added to the paragraph 'monitoring of surgery during the trial', page 9:

"Surgeons will be encouraged to submit photos for each patient recruited to the ROMIO study, to demonstrate that the key operative components have been performed."

3. The protocol should describe how the data from the QA measures will be analysed and used to develop and validate a set of QA measures for future pragmatic trials and also describe how agreement will be reached on the use of such measures in future pragmatic trials.

Reply: We thank the reviewer for this comment and agree that these issues are extremely important. However, we will not be in a position to write an analysis plan until the QA measures have been developed. This will be the focus of a future paper. Regarding obtaining agreement about use of these measures in future pragmatic trials, this is again something that will be possible to describe in more detail once the measures have been developed. Moreover, it is the focus of a current grant which aims to establish how best to obtain consensus on QA measures in surgical trials.

Revisions: We have added a paragraph to the end of the manuscript entitled 'future work', which outlines potential analysis plans:

"Future work

This study aims to develop methods for measuring the QA of surgical interventions. Once developed, they will be implemented in the context of an RCT comparing open and laparoscopic surgery for patients with oesophageal cancer. There are numerous analyses that may be undertaken, which will depend on the exact nature of the QA measures that are developed. First, the quality of intervention delivery will be assessed, and to compare what surgeons reported in the case report forms (i.e. what they said they did) with the intra-operative photographs (i.e. what actually happened). Second, we will explore trends relating to surgeon skill (from the operative videos) and patient outcomes, though numbers are relatively small and this may not, therefore, be possible. Third, the process of developing QA measures for this study will influence future work in this area; specifically, the generation of guidance that can be extrapolated to other RCTs in surgery. This is the focus of a funded fellowship award, which will also examine the process of obtaining consensus about exactly what these QA measures should comprise, accounting for trial design and the nature of the interventions under investigation. Finally, it is important to recognise that improving the QA of surgical interventions may improve standards of surgery within an RCT. Whilst we will document the points at which feedback is given to surgeons and centres, it may not be realistically possible to correlate this (and assign causation) with outcomes. It is well recognised that patients within RCTs generally have more

favourable outcomes than those that do not. Improvements to the standards of intervention delivery may, therefore, form a part of the benefits associated with trial participation.”

VERSION 2 – REVIEW

REVIEWER	Lawrence Kim University of North Carolina, USA
REVIEW RETURNED	16-Dec-2018

GENERAL COMMENTS	The authors have thoughtfully responded to the criticisms. While I do not necessarily agree with all of their assertions, their points are rational and justified.
--

REVIEWER	Paco Welsing UMC Utrecht, The Netherlands
REVIEW RETURNED	13-Dec-2018

GENERAL COMMENTS	I am satisfied with the answers to my queries and have no further comments.
---